# Lightweight Robust Direct Preference Optimization

**Cheol Woo Kim**[*]   **Shresth Verma**[*]   **Mauricio Tec**[*]   **Milind Tambe**
School of Engineering and Applied Sciences, Harvard University, Boston, MA 02134
{cwkim, sverma, mauriciogtec, tambe}@g.harvard.edu

## Abstract

Direct Preference Optimization (DPO) has become a popular method for fine-tuning large language models (LLMs) due to its stability and simplicity. However, it is also known to be sensitive to noise in the data and prone to overfitting. Recent works have proposed using distributionally robust optimization (DRO) to address potential noise and distributional shift in the data. However, these methods often suffer from excessive conservatism and high computational cost. We propose DPO-PRO (**DPO** with **P**reference **Ro**bustness), a robust fine-tuning algorithm based on DPO which accounts for uncertainty in the preference distribution through a lightweight DRO formulation. Unlike prior DRO-based variants, DPO-PRO focuses solely on uncertainty in preferences, avoiding unnecessary conservatism and incurring negligible computational overhead. We further show that DPO-PRO is equivalent to a regularized DPO objective that penalizes model overconfidence under weak preference signals. We evaluate DPO-PRO on standard alignment benchmarks and a real-world public health task. Experimental results show that our method consistently improves robustness to noisy preference signals compared to existing DPO variants.

## 1   Introduction

Reinforcement learning from human feedback (RLHF) enables training complex reinforcement learning (RL) policies without manually specifying reward functions. A common approach is to first learn a reward function from a preference data data using maximum likelihood estimation. This learned reward is then used to define an RL objective, typically augmented with a KL-penalty term, and optimized using algorithms such as Proximal Policy Optimization (PPO) [31]. RLHF methods have been successful in fine-tuning large language models (LLMs) to better align with human preferences [13, 5, 27].

A popular alternative to the standard two-stage pipeline is to directly optimize the policy using offline preference data, an approach known as Direct Preference Optimization (DPO) [29]. DPO is generally known to be more stable and easier to train than PPO. However, it is also known to be vulnerable to noise in the training data and prone to overfitting [16, 2, 41, 37]. This issue is further compounded by the growing use of LLM-annotated [44, 32, 14, 45] or reward model–generated [18, 15, 22, 19] preference data in recent work, which may deviate from actual human preferences.

To address the noise and distributional shifts associated with training data, distributionally robust optimization (DRO) has been applied to DPO [36, 40, 24]. DRO [30] optimizes the worst-case expected loss over a predefined set of distributions. While this approach can provide robustness, it can by overly conservative and introduces computational difficulties, as the resulting min–max robust objective for DPO is often intractable.

---

[*]Equal Contribution

39th Conference on Neural Information Processing Systems (NeurIPS 2025) Workshop: Reliable ML from Unreliable Data.

To tackle these challenges, we introduce DPO-PRO (**DPO** with **P**reference **Ro**bustness), a lightweight and distributionally robust enhancement of DPO [29]. DPO-PRO incorporates uncertainty in the preference distribution using an efficient yet effective DRO formulation [30], ensuring that the fine-tuned policy remains reliable under imperfect preference annotation. Unlike prior DRO-based DPO methods, our approach focuses solely on robustness to the preference distribution, rather than the full data-generating process. This avoids excessive conservatism and introduces only negligible computational overhead. We further show that the resulting robust loss can be interpreted as a regularized DPO loss that penalizes model overconfidence and weak preference signals.

We evaluate our approach on a range of publicly available standard benchmarks, as well as a real-world public health–related task for LLMs. Our experiments show that DPO-PRO is more robust to noisy preference supervision than both vanilla DPO and prior DRO-based methods.

## 2 Related Works

A growing body of work has explored robustness in RLHF and DPO, particularly by leveraging DRO. [36] introduce a DRO-based DPO formulation using a KL-divergence-based ambiguity set. [40] extend this approach to both KL and Wasserstein distances, while [24] explore robustness under total variation distance, applying it to both DPO and standard two-stage RLHF.

In contrast to these DRO-based approaches, other works assume explicit corruption models. For example, [12] assume that preference labels are flipped according to a known noise rate, and [9] model preferences using a Bradley–Terry (BT) framework [8] with added noise on the reward differences.

[20] use a DRO framework to develop adaptive loss functions for reward model learning, but focus is not on distributional robustness. [43] introduce robustness by constructing confidence sets over learned reward functions, followed by pessimistic policy optimization.

Broadly, existing robustness techniques fall into two categories: (1) methods relying on strong assumptions about noise models, which may be unrealistic in practice, and (2) DRO-based methods that avoid explicit corruption modeling but hedge against worst-case distributions. As discussed in Section 3.2, standard DRO approaches tend to be overly conservative and often require solving challenging min–max optimization problems. For tractability, prior methods rely on heuristic approximations [36, 40], potentially undermining theoretical guarantees or practical robustness. In contrast, DPO-PRO is compatible with standard gradient-based optimization without relying on heuristics, making it practically efficient. It also avoids unnecessary conservatism and still provides robustness guarantees to preference signals.

## 3 Preliminaries

### 3.1 Data Distribution

In DPO, each data point is a tuple $(x, y_1, y_2, c)$. A prompt $x \in \mathcal{X}$ is drawn from a distribution $\mu$. Given $x$, responses $y_1, y_2 \in \mathcal{Y}$ are sampled independently from a policy $\pi(\cdot|x)$. A label $c \in \{1, -1\}$ is a Bernoulli random variable indicating whether response $y_1$ is preferred over $y_2$, with preference probability given by $p^*(y_1 \succ y_2|x)$. The distribution $p^*$ represents the ground-truth preference distribution that the user intends the language model to align with.

In summary, the data-generating process involves three components: (1) the prompt distribution, (2) the response distribution, and (3) the preference distribution. This yields the following joint distribution: $P(x, y_1, y_2, c) = \mu(x)\pi(y_1|x)\pi(y_2|x)\left[\mathbb{I}_{\{c=1\}}\, p^*(y_1 \succ y_2 \mid x) + \mathbb{I}_{\{c=-1\}}\, p^*(y_2 \succ y_1 \mid x)\right]$. We often omit the dependence of $p^*$ (and other preference distributions) on $(x, y_1, y_2)$ and simply write $p^*$ instead of $p^*(y_1 \succ y_2|x)$ when the meaning is clear from context.

### 3.2 (Distributionally Robust) DPO

In standard RLHF, the first step is to learn a reward function $R_\phi$ from a fixed dataset $\mathcal{D}$. This is typically done by minimizing the negative log-likelihood:

$$-\mathbb{E}_{(x, y_1, y_2, c) \sim \mathcal{D}}\left[\log \sigma\left(c \cdot (R_\phi(x, y_1) - R_\phi(x, y_2))\right)\right],$$

where $\sigma$ is the sigmoid function.

Once the reward model is trained, a policy $\pi_\theta$ is optimized to maximize the expected reward while remaining close to a reference policy $\pi_{ref}$, typically via PPO [31]:

$$\max_{\pi_\theta} \mathbb{E}_{(x,y)\sim\pi_\theta} [R_\phi(x,y)] - \beta \, \mathrm{KL}(\pi_\theta \, \| \, \pi_{\mathrm{ref}}),$$

where $\beta$ is a regularization parameter controlling the strength of the KL penalty.

DPO simplifies this pipeline by combining the two stages into a single objective that directly optimizes the policy from pairwise preference data. The DPO loss $\mathcal{L}_{\mathrm{DPO}}(\pi_\theta)$ is given by:

$$-\mathbb{E}_{(x,y_1,y_2,c)\sim\mathcal{D}}\Big[\log \sigma\big(c\,\beta\big(\log \tfrac{\pi_\theta(y_1|x)}{\pi_{\mathrm{ref}}(y_1|x)} - \log \tfrac{\pi_\theta(y_2|x)}{\pi_{\mathrm{ref}}(y_2|x)}\big)\big)\Big].$$

For brevity, we define the per-sample loss as:

$$\ell_\theta(x,y_1,y_2,c) = -\log \sigma\Big(c\,\beta\Big(\log \tfrac{\pi_\theta(y_1|x)}{\pi_{\mathrm{ref}}(y_1|x)} - \log \tfrac{\pi_\theta(y_2|x)}{\pi_{\mathrm{ref}}(y_2|x)}\Big)\Big).$$

Using this expression, $\mathcal{L}_{\mathrm{DPO}}(\pi_\theta)$ is also equivalent to:

$$\mathbb{E}_{(x,y_1,y_2)\sim\mathcal{D}}[p^*\ell_\theta(x,y_1,y_2,1) + (1-p^*)\ell_\theta(x,y_1,y_2,-1)].$$

When the context is clear that the triplet $(x,y_1,y_2)$ and the parameter $\theta$ is fixed, we also use $\ell_1$ and $\ell_{-1}$ to denote $\ell_\theta(x,y_1,y_2,1)$ and $\ell_\theta(x,y_1,y_2,-1)$, respectively.

As discussed earlier, DPO is vulnerable to overfitting and noise in the data. To address this, recent work has proposed distributionally robust objectives of the form $\max_{P\in\mathcal{Q}(\mathcal{D})} \mathcal{L}_P(\pi_\theta)$, where $\mathcal{Q}(\mathcal{D})$ denotes an ambiguity set centered around the empirical distribution (data) $\mathcal{D}$ and the loss $\mathcal{L}_P$ is the DPO loss evaluated under the perturbed distribution $P$ rather than the original data distribution [40, 36].

**Conservatism and Computational Costs**  The above DRO formulation hedges against shifts in the entire joint distribution on $(x,y_1,y_2,c)$. This allows the adversary to assign weights to highly unlikely prompts or responses. The outer minimization must then optimize against losses in these practically irrelevant regions, which might make the update overly conservative and slow improvement on the data the model actually sees. From a computational standpoint, solving the resulting min–max problem can be intractable in practice, and existing methods often rely on approximations that deviate from the original DRO formulation.

## 4   DPO with Preference Robustness

In this section, we introduce DPO-PRO, a distributionally robust version of DPO that specifically targets distributional shifts or other forms of noise in the preference distribution.

### 4.1   Uncertainties in the Preference Distribution

In each data point $(x,y_1,y_2,c)$, the triplet $(x,y_1,y_2)$ represents observed content, which typically comes from a well-understood and controllable data collection process. In general, $x$ is drawn from a pool of candidate inputs, and the response pair $(y_1,y_2)$ is generated by the reference policy $\pi_{ref}$ (usually a supervised fine-tuned model) for every sampled prompt $x$. As more data is collected, the uncertainty in this part of the data is expected to diminish. Noise in the prompts and responses becomes even less of an issue under iterative DPO methods, which are increasingly adopted in practice [39, 11, 38], as the prompt and response pairs are continually updated.

In contrast, we argue that the preference distribution $c \sim p^*$ is inherently noisy in practice, and collecting more data is unlikely to resolve the underlying uncertainty in human preference. Noise can arise from multiple sources. For example, errors can be made by LLM judges or reward models when these proxies are used for annotation. Even when humans provide annotations directly, other sources of noise remain, including human subjectivity, irrational or inconsistent behavior, and temporal variability (e.g., the same annotator providing different judgments at different times). This noise is persistent and cannot be easily addressed through better data curation or increased data volume. In

other words, the ground-truth preference distribution $p^*$ is an idealized and inaccessible object in practice. Therefore, we argue that uncertainty in the preference distribution should be the primary motivation for incorporating robustness into DPO.

We assume that for each prompt and response pair $(x, y_1, y_2)$, we have access to a (potentially noisy) preference distribution $q(y_1 \succ y_2|x)$. We apply DRO adjustment, assuming worst-case deviation from $q$ within a pre-specified chi-squared divergence ball. Formally, we define the DRO loss $\mathcal{L}_{\text{DRO}}(\pi_\theta)$ as:

$$\mathbb{E}_{(x,y_1,y_2)\sim\mathcal{D}} \max_{p\in Q(x,y_1,y_2,\rho)} \mathbb{E}_{c\sim p}\Big[\ell_\theta(x, y_1, y_2, c)\Big], \tag{1}$$

where $Q(x, y_1, y_2\rho) = \big\{p : \chi^2\big(p \,\|\, q(y_1 \succ y_2 \mid x)\big) \leq \rho\big\}$. This formulation also resembles a zero-sum Stackelberg game, as noted in prior work [35, 3], where the optimizer (leader) selects $\theta$, and an adversary (follower) chooses a worst-case distribution in response. The goal is to find the optimal $\theta$, anticipating that the follower will act adversarially to maximize the loss.

Note that our formulation avoids unnecessary robustification over the joint distribution on $(x, y_1, y_2)$, and assumes that this part of the data is reliable (i.e., the adversary does not perturb this part of the distribution). Furthermore, unlike standard DRO methods that can add significant computational burden [30], our formulation introduces negligible additional cost as shown next.

### 4.2 Computing Worst-case Distribution

Consider a single data point $(x, y_1, y_2)$. We assume access to an estimate of the probability that one response is preferred over the other: $q(y_1 \succ y_2|x) \in (0, 1)$. Given this probability $q$, we define the following per-sample optimization problem:

$$\max_{p\in[0,1]} \quad p\,\ell_\theta(x, y_1, y_2, 1) + (1 - p)\,\ell_\theta(x, y_1, y_2, -1)$$

$$\text{s.t.} \quad \frac{(p - q)^2}{q(1 - q)} \,\leq\, \rho \tag{2}$$

This problem finds the worst-case distribution within the chi-squared ambiguity set centered at $q$, $\Big\{p : \chi^2\big(p \,\|\, q(y_1 \succ y_2 \mid x)\big) \equiv \frac{(p-q)^2}{q(1-q)} \leq \rho\Big\}$. Let $\hat{p}(y_1 \succ y_2|x)$ denote the optimal solution to this problem. Using this, the robust loss $\mathcal{L}_{\text{DRO}}(\pi_\theta)$ can be equivalently expressed as

$$\mathbb{E}_{(x,y_1,y_2)\sim\mathcal{D}}\Big[\hat{p}\,\ell_\theta(x, y_1, y_2, 1) + (1 - \hat{p})\,\ell_\theta(x, y_1, y_2, -1)\Big] \tag{3}$$

Problem (2) is a one-dimensional optimization problem with a linear objective, which admits a simple closed-form solution:

$$\hat{p}(y_1 \succ y_2|x) = \begin{cases} \min\Big\{1, q + \sqrt{\rho\,q(1 - q)}\Big\}, & \text{if } \ell_1 \geq \ell_{-1}, \\ \max\Big\{0, q - \sqrt{\rho\,q(1 - q)}\Big\}, & \text{if } \ell_1 < \ell_{-1}. \end{cases}$$

This closed-form solution can be directly substituted into the loss in Eq (3), resulting in a fully closed-form expression for the DRO-adjusted loss.

We now explain the intuition behind the closed-from worst-case distribution $\hat{p}$. Consider the case where $\ell_1 \geq \ell_{-1}$, which implies that under the current policy $\theta$, the model assigns a higher likelihood to $y_2$ being preferred over $y_1$. To increase the loss, the adversary seeks to shift the preference probability $q(y_1 \succ y_2 \mid x)$ in the opposite direction. That is, it tries to increase $q$ to emphasize a preference for $y_1$, which contradicts the model's belief. Consequently, the worst-case distribution becomes $q + \sqrt{\rho\,q(1 - q)}$. Since probabilities must remain within the unit interval, we clip the value at 1, yielding the final expression $\min\Big\{1, \, q + \sqrt{\rho\,q(1 - q)}\Big\}$.

### 4.3 Efficient Optimization of the Loss

The worst-case preference probability $\hat{p}$, obtained in closed-form above, can be directly substituted into the per-sample DPO gradient to compute the gradient for DPO-PRO:

$$\hat{p}\,\nabla_\theta\ell_\theta(x, y_1, y_2, 1) \,+\, (1 - \hat{p})\,\nabla_\theta\ell_\theta(x, y_1, y_2, -1). \tag{4}$$

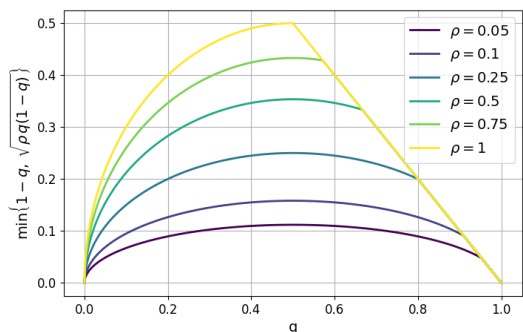

Figure 1: Visualization of the uncertainty-weighted coefficient for various values of $\rho$. This coefficient attains its maximum near $q = 0.5$ and decreases as $q \to 0$ or $q \to 1$. For small values of $\rho$, the maximum is achieved exactly at $q = 0.5$, where $\sqrt{\rho\, q(1-q)}$ peaks. However, for larger $\rho$, the maximum occurs at the intersection point between the curves $1 - q$ and $\sqrt{\rho\, q(1-q)}$. This is because $q + \sqrt{\rho\, q(1-q)}$ (the worst-case distribution under the chi-squared divergence constraint) may exceed 1 when $\rho$ is large, causing the worst-case distribution $\hat{p}$ to be clipped at 1. In such cases, the adversary's perturbation $\hat{p} - q$ can be greater when $q < 0.5$, leading the penalty term to peak at a value of $q$ smaller than 0.5.

**Proposition 4.1.** *Eq* (4) *provides an unbiased gradient estimate of the* DPO-PRO *loss in Eq* (1).

Note that we do not formally differentiate through the inner maximization in Eq (1) with respect to $\theta$. However, Danskin's theorem [7] justifies our approach. For a fixed $\theta$, we may solve the inner maximization and directly substitute the resulting worst-case $\hat{p}$ into the gradient expression.

We emphasize the resulting DRO gradient is both exact and computationally efficient. Unlike prior work, our approach does not introduce any approximation or heuristic in minimizing the DRO objective. Moreover, our distributional robustness is applied only to the preference distribution $q$, rather than the full data-generating distribution on $(x, y_1, y_2, c)$. Hence, the resulting method is significantly less conservative than earlier DRO-based DPO approaches.

### 4.4 Analysis of the Loss Function

In this section, we analyze the robust loss $\mathcal{L}_{\text{DRO}}$ and show that it is equivalent to the original DPO loss augmented with a regularization term. This interpretation provides insight into how the DRO formulation affects model learning.

**Proposition 4.2.** *The DRO loss $\mathcal{L}_{DRO}$ is equivalent to regularizing the original DPO loss $\mathcal{L}_{DPO}$ as:*

$$\mathcal{L}_{DRO} = \begin{cases} \mathcal{L}_{DPO} + \min\left\{1 - q,\ \sqrt{\rho\, q(1-q)}\right\}(\ell_1 - \ell_{-1}), & \text{if } \ell_1 \geq \ell_{-1}, \\ \mathcal{L}_{DPO} + \min\left\{q,\ \sqrt{\rho\, q(1-q)}\right\}(\ell_{-1} - \ell_1), & \text{if } \ell_1 < \ell_{-1}. \end{cases}$$

We now analyze the behavior of the regularization term, assuming that $\ell_1 \geq \ell_{-1}$. The opposite case follows symmetric logic.

**(1) Uncertainty-Weighted Coefficient.** The first factor $\min\left\{1 - q,\ \sqrt{\rho\, q(1-q)}\right\}$ penalizes cases where the preference signal is uncertain (i.e., when $q$ is close to 0.5). This term is largest when $q$ is near 0.5 and diminishes as $q \to 0$ or $q \to 1$. Its magnitude is modulated by the robustness radius $\rho$, with larger $\rho$ leading to a stronger penalty. We visualize this uncertainty-weighted coefficient for various $\rho$ values in Figure 1.

**(2) Model Confidence.** The second factor $(\ell_1 - \ell_{-1})$ reflects the log-odds of the current model's preference for $y_2$ over $y_1$. A large value indicates strong model confidence. This term increases when the model becomes more certain about the preference, regardless of the ground-truth label.

**(3) Combined Effect.** The regularization term penalizes the model for being overly confident in its preferences when the preference signal $q$ is ambiguous. In contrast, when the signal is strong (i.e., $q$ is close to 0 or 1), the penalty diminishes, allowing the model to express stronger preferences. In effect, the DRO loss encourages calibrated learning: the model is allowed to be confident only when the preference signal is also confident.

### 4.5 Obtaining the Soft Score $q$ in Practice

The DRO adjustment described above requires a soft score $q(y_1 \succ y_2 \mid x)$, rather than a binary annotation for each prompt and response pair $(x, y_1, y_2)$, which is the typical format in preference learning. Fortunately, such soft scores are increasingly available in modern DPO pipelines.

If the dataset contains multiple binary annotations for the same pair (e.g., from different annotators), $q(y_1 \succ y_2 \mid x)$ can be estimated by averaging the votes. When LLM judges are used, as in [44, 32, 14, 45], the preference score can be obtained in several ways: (1) producing multiple binary judgments through repeated queries and averaging the outcomes; (2) prompting the model to directly output a numerical preference score; or (3) extracting log-probabilities assigned to preference tokens (e.g., `1` for $y_1 \succ y_2$, `-1` for $y_2 \succ y_1$), followed by a softmax transformation to estimate a distribution over preferences [21]. Similarly, when an off-the-shelf reward model is used [18, 15, 22, 19], its scalar outputs can be converted into pairwise preference probabilities using the BT model.

In many practical settings, particularly those involving expensive human annotation, only a binary label may be available per $(x, y_1, y_2)$. In this case, we observe only one sample from the underlying Bernoulli distribution $q(y_1 \succ y_2 \mid x)$, making estimation of $q$ difficult. Additionally, KL divergence becomes ill-defined when $q$ is exactly 0 or 1 due to a zero denominator. To address this, we slightly relax the chi-squared divergence constraint and rewrite it as $(p - q)^2 \leq \rho\, q(1 - q)$, which reduces to $\hat{p} = q$ when $q \in \{0, 1\}$. As a result, when binary labels are used, the DRO loss reduces exactly to the standard DPO loss, introducing no additional regularization. This naturally unifies the treatment of binary and soft labels. Only soft annotations (such as those from LLM judges, reward models, or aggregated human votes) contribute regularization, while binary labels are treated as ground truth.

If additional robustness is desired even under binary supervision, one can apply smoothing by replacing label 1 with $1 - \epsilon$, and 0 with $\epsilon$, where $\epsilon$ is a small hyperparameter. The DRO formulation can then be applied to this smoothed label. This makes the uncertainty-weighted coefficient uniform across all pairs (since $q$ only takes values $1 - \epsilon$ or $\epsilon$), but the $(\ell_1 - \ell_{-1})$ term still encourages the model to reduce overconfidence in its current preference.

## 5 Experiments

We evaluate DPO-PRO in two settings. First, we validate the approach on a standardized benchmark for preference-based LLM fine-tuning. Second, we evaluate the method on a reward function design task for a real-world public health resource allocation problem.

### 5.1 Benchmarks on General Alignment Datasets

We use the recent UltraFeedback alignment benchmark dataset [14], consisting of 60,000 top-quality human pairwise preference annotations that cover a wide variety of tasks and instruction types, rendering it a broad benchmark for examining alignment research.

**Noise in the training data** Our primary objective is to evaluate how effectively preference-based LLM fine-tuning methods handle shifts in preference distributions between training and evaluation data. We use $q^*$ denote the true preference distribution used at test time. For our experiments, we define $q^*$ using the state-of-the-art preference reward model Eurus 7B[42], which has been trained for the UltraFeedback dataset [14]. We compute $q^*$ using the BT model.

To simulate noise during training, we construct a noisy preference distribution $q_\alpha$ for training as the mixture

$$q_\alpha := q^*(1 - \alpha) + (1 - q^*)\alpha. \tag{5}$$

Intuitively, $q_\alpha$ "flips" the preference distribution with probability $\alpha$. We refer to this type of noise interchangeably as *label switching*. During training, we have access to $q_\alpha$ but not to $q^*$, with $\alpha$ being unknown.

**Training details**   We base our experiments on the recent Phi 3-mini 3B model without instruction [1]. We initially train for two epochs of supervised fine tuning (SFT), followed by one epoch of DPO, with the baselines discussed below. All experiments are performed on a Nvidia A100 GPU 40 GB with batch size 2. Refer to the technical appendix for full details on the training procedure.

**Baselines**   We compare DPO-PRO against the standard DPO [29] and the prior distributionally robust variant DrDPO [36]. We evaluate DPO-PRO under several values of the robustness parameter $\rho$ based on the $\chi^2$-divergence. Models are trained under two levels of label-flip noise, $\alpha = 0.3$ (low) and $\alpha = 0.6$ (high), as well as the noiseless setting $\alpha = 0$.

**Evaluation metrics**   After the fine-tuning step, we generate a response $y_g \sim \pi_\theta(x)$ for every prompt $x$ in the evaluation dataset and compute its reward $R_\phi(x, y_g)$. We consider three evaluation metrics. The first is the *win rate*, computed by comparing the generated response with the chosen response $y_c$ in the dataset as $\texttt{win\_rate} = \frac{1}{N_{\text{eval}}} \sum_{i=1}^{N_{\text{eval}}} \mathbb{I}\big(R_\phi(x, y_g) > R_\phi(x, y_c)\big)$, where $\mathbb{I}$ is the indicator function. The second metric, *generation evaluation reward*, is defined as $\texttt{eval\_reward} = \frac{1}{N_{\text{eval}}} \sum_{i=1}^{N_{\text{eval}}} R_\phi(x, y_g)$. Finally, we also evaluate responses using an LLM-as-a-judge approach with GPT-4o-mini [26]. To mitigate position bias, we randomly flip the order of the two responses presented to the judge.

**Results**   Table 1 summarizes the results. For metrics based on reward models, DPO-PRO outperforms the baselines across all noise levels, including the noiseless case. As expected, performance declines as noise increases for all methods. The effect of the robustness parameter $\rho$ is also consistent with intuition: smaller $\rho$ values tend to perform better under low noise but exhibit reduced robustness under higher noise, whereas larger $\rho$ values maintain more stable performance.

For the LLM-as-a-judge evaluation, DPO-PRO generally underperforms when noise is small but improves relative to other methods as the noise level increases. Under high noise, DPO-PRO surpasses both vanilla DPO and DrDPO, reflecting the same robustness trend observed with the reward-based metrics.

## 5.2   Reward Function Design Task

Recently, several works explored using LLMs to automate reward function design for RL [25, 17, 10, 23, 6, 33]. In these settings, the LLM is tasked with interpreting a user's high-level objective provided as a natural language prompt (e.g., description of desired policy outcomes), and directly generating a complete reward function that guides RL policy training.

We apply this task to the context of restless multi-armed bandits (RMABs), a class of Markov decision processes designed for sequential decision-making under resource constraints. Specifically, we focus on a public health application of RMABs for allocating limited resources across population subgroups based on various features such as demographic information. Here, decision-makers must encode their priorities (e.g., which populations to prioritize) into a reward function so that the resulting policy aligns with long-term public health goals.

Our experiments use real-world data from a maternal mobile health program operated by ARMMAN [4], a nonprofit organization that delivers preventive care messages via phone calls. In our simulated environment based on this data, health workers can place a limited number of live service calls each week to improve beneficiary engagement with the program. Additional details on the RMAB formulation and the public health setting are provided in the Appendix.

In this setting, the prompt $x$ represents a desired policy outcome specified by a human prompter, for example, `Prioritize young and low-income beneficiaries`. The responses $y_1$ and $y_2$ correspond to candidate reward functions for RMABs. Each reward function takes as input features including the beneficiary's demographic information and engagement status, which is represented as a binary variable, and produces a scalar reward.

Table 1: Results on UltraFeedback under label-flip noise.

(a) Win rate (%) based on reward model (↑).

| Method | No noise ($\alpha = 0$) | Low noise ($\alpha = 0.3$) | High noise ($\alpha = 0.6$) |
|---|---|---|---|
| DPO-PRO ($\rho = 0.008$) | 32.56 | 27.76 | 22.02 |
| DPO-PRO ($\rho = 0.03$) | 31.31 | 27.70 | 23.21 |
| DPO-PRO ($\rho = 0.1$) | 27.20 | 27.04 | 24.06 |
| DPO | 29.80 | 28.23 | 21.77 |
| DrDPO | 29.40 | 29.14 | 21.36 |

(b) Average evaluation reward (↑).

| Method | No noise ($\alpha = 0$) | Low noise ($\alpha = 0.3$) | High noise ($\alpha = 0.6$) |
|---|---|---|---|
| DPO-PRO ($\rho = 0.008$) | 923.70 | 713.00 | 446.65 |
| DPO-PRO ($\rho = 0.03$) | 912.65 | 695.00 | 499.59 |
| DPO-PRO ($\rho = 0.1$) | 781.28 | 656.32 | 499.36 |
| DPO | 877.01 | 729.12 | 400.44 |
| DrDPO | 897.67 | 753.77 | 419.05 |

(c) Win rate (%) based on LLM-as-a-judge evaluation (↑).

| Method | No noise ($\alpha = 0$) | Low noise ($\alpha = 0.3$) | High noise ($\alpha = 0.6$) |
|---|---|---|---|
| DPO-PRO ($\rho = 0.008$) | 43.24 | 35.17 | 28.75 |
| DPO-PRO ($\rho = 0.03$) | 41.71 | 33.33 | 28.98 |
| DPO-PRO ($\rho = 0.1$) | 40.33 | 33.54 | 27.97 |
| DPO | 43.77 | 36.19 | 26.97 |
| DrDPO | 44.85 | 38.84 | 25.51 |

A key challenge in this task is that the quality of a reward function cannot be assessed in isolation. Rather, it must be evaluated based on the policy it induces when optimized, i.e., how effectively the resulting policy fulfills the prompter's (often ambiguous) intent. This requires the annotator to infer the downstream effects of each reward function. Given the complexity and ambiguity involved, the resulting preference signals are inevitably noisy and should not be treated as fully reliable.

**Preference Dataset Construction**  We generate a preference dataset of health worker prioritization commands, preferred, and rejected reward functions. Since it is costly to generate this dataset through human annotation, we use an LLM judge with ChatGPT 4o-mini [26] as follows: i) query LLM-judge to obtain 20 candidate reward functions that align with the prioritization command, ii) for each candidate reward function, solve RMAB problem using Whittle index method [34] and generate trajectory outputs (see the Appendix for the explanation on the Whittle index), iii) sample 50 pairs of reward functions from this set and query the LLM-judge to select preferred reward function, iv) perform this query 10 times to estimate the LLM's uncertainty over preferences. Finally, we obtain a dataset of 9500 preferred and rejected reward function responses over 190 prioritization prompts (see Appendix for more details).

**Noise injection, training details, and baselines**  We adopt the same noise injection scheme to obtain $q_\alpha$ as described in Section 5.1 and Eq (5). We fine-tune a Llama 3 (8B) model, keeping the learning rate, DPO reward temperature, and other hyperparameters identical to those used in the UltraFeedback benchmark experiments. We fix $\rho = 0.1$ in this experiment.

**Evaluation**  To evaluate the quality of a generated reward function, we follow a procedure analogous to the dataset creation: i) solve the resulting RMAB problem using the Whittle Index method to generate policy trajectories, ii) use LLM judge (GPT-4o-mini) to evaluate these policy outcomes. Specifically, we compute the win rate by comparing the policy induced by our generated reward function to the policy obtained from the reward function in the training data.

**Results**  Table 2 shows the results. In zero-noise setting, vanilla-DPO outperforms robust methods. However, as noise level increases, DPO-PRO outperforms vanilla DPO and DrDPO.

Table 2: Win rate (%) based on LLM-as-a-judge evaluation for the reward function design task (↑).

| Noise Setting | DPO | DrDPO | DPO-PRO |
|---|---|---|---|
| No noise ($\alpha = 0$) | 55.0 | 40.0 | 35.3 |
| Low noise ($\alpha = 0.3$) | 45.0 | 54.8 | 63.9 |

Table 3: Examples of reward functions produced by vanilla-DPO and DPO-PRO under noisy training conditions. `s` indicates binary engagement status, and other input features (e.g., demographic attributes or call timing) are also encoded as binary variables. DPO-PRO produces more conservative interpretations of preferences.

| **Task 1:** Prefer both young and elderly beneficiaries | |
|---|---|
| Vanilla-DPO | `s + 3 * (youngest_age or second_youngest_age or oldest_age) + 2 * (lowest_education or second_lowest_education or third_lowest_education) + (lowest_income or second_lowest_income or third_lowest_income)` |
| DPO-PRO | `s + 3 * (youngest_age or second_youngest_age or oldest_age) + 2 * (lowest_education or second_lowest_education or third_lowest_education)` |

| **Task 2:** Prioritize midday calls and NGO-registered users | |
|---|---|
| Vanilla-DPO | `s + 3 * (10_30-12_30pm and NGO_registered) + 2 * (12_30-3pm and NGO_registered)` |
| DPO-PRO | `s + 3 * (12_30-3pm and NGO_registered)` |

To further illustrate the differences between vanilla DPO and DPO-PRO, we present sample response outputs from both models on two representative tasks from the evaluation dataset (Table 3). Many tasks contain attributes that can be interpreted in multiple ways. For instance, the concept of "young beneficiaries" can be identified directly through the age feature, but also indirectly via lower education level or income. Similarly, a task that expresses a preference for "midday calls" could refer explicitly to the 12:30–3:00 PM time slot, but might also include the 10:30–12:30 PM window, depending on interpretation.

In such cases, we observe that vanilla-DPO favors uncommon interpretations of ambiguous attributes based on spurious correlations in noisy preference data. In contrast, DPO-PRO adopts a cautious strategy by selecting narrower definition, aiming to be robust to uncertainty in the preference signal at evaluation.

## 6 Conclusion

We introduced DPO-PRO, a robust fine-tuning algorithm that extends DPO by explicitly modeling uncertainty in preference distributions using a lightweight DRO formulation. Compared to existing DRO-based methods, DPO-PRO avoids excessive conservatism while offering a strong theoretical justification as a regularized variant of DPO. Through evaluations on standard alignment benchmarks and a reward function design task in public health, we showed that DPO-PRO improves robustness to noisy preference signals over existing DPO variants.

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

# A  Codebase

We release the codebase for all our experiments on finetuning LLMs here: https://anonymous.4open.science/r/DPO-Pro-A33D/README.md

# B  Extensions and Generalizations

## B.1  Beyond the Chi-squared Divergence

We use the chi-squared divergence to define the ambiguity set due to the simplicity of its resulting closed-form solution. However, alternative divergence measures can also be used and are likely to offer comparable computational efficiency as ours. This is because the per-sample optimization problem in Eq (2) remains a one-dimensional problem with linear objective. For example, using the KL divergence leads to a constraint of the form: $p \log \frac{p}{q} + (1-p) \log \frac{1-p}{1-q} \leq \rho$. In this case, the worst-case $\hat{p}$ can be computed by solving the equation $p \log \frac{p}{q} + (1-p) \log \frac{1-p}{1-q} = \rho$, which can be efficiently handled using a simple one-dimensional root-finding method.

# C  Omitted Proofs

## C.1  Proof of Proposition 4.1

The result follows from Danskin's theorem [7]. To apply the theorem, we must verify that the inner maximization problem in Eq. (2) has a compact feasible set and admits a unique optimal solution.

To establish compactness, we need to show that the feasible set is both bounded and closed. Boundedness is immediate since $p \in [0, 1]$. Closedness follows from the fact that the constraint function $p \mapsto (p - q)^2 - \rho q(1 - q)$ is continous in $p$. Hence, the set $(p - q)^2 \leq \rho q(1 - q)$ is closed. Since $p \in [0, 1]$ is also closed and the intersection of closed sets is closed, the feasible set is closed. Uniqueness of the optimal solution $\hat{p}$ is also straightforward, since the objective is one-dimensional linear function in $p$.

## C.2  Proof of Proposition 4.2

We begin by assuming the case $\ell_1 \geq \ell_{-1}$. Then,

$$
\begin{aligned}
\mathcal{L}_{\text{DRO}} &= \hat{p}\, \ell_1 + (1 - \hat{p})\, \ell_{-1} \\
&= q\, \ell_1 + (1 - q)\, \ell_{-1} + (\hat{p} - q)(\ell_1 - \ell_{-1}) \\
&= \mathcal{L}_{\text{DPO}} + \underbrace{\min\left\{1 - q,\ \sqrt{\rho\, q(1 - q)}\right\}}_{\text{Uncertainty-Weighted Coefficient}}(\ell_1 - \ell_{-1}).
\end{aligned}
$$

For the opposite case, when $\ell_{-1} > \ell_1$, the derivation proceeds symmetrically.

$$
\begin{aligned}
\mathcal{L}_{\text{DRO}} &= \hat{p}\, \ell_1 + (1 - \hat{p})\, \ell_{-1} \\
&= \mathcal{L}_{\text{DPO}} + (\hat{p} - q)(\ell_1 - \ell_{-1}) \\
&= \mathcal{L}_{\text{DPO}} - \min\left\{q,\ \sqrt{\rho\, q(1 - q)}\right\}(\ell_1 - \ell_{-1}) \\
&= \mathcal{L}_{\text{DPO}} + \min\left\{q,\ \sqrt{\rho\, q(1 - q)}\right\}(\ell_{-1} - \ell_1).
\end{aligned}
$$

# D  Experimental Details

## D.1  Description of the Baseline

DrDPO applies DRO over the empirical data distribution using a KL-divergence-based ambiguity set:

$$
\min_{\pi_\theta} \max_{P \in \mathcal{Q}(\mathcal{D})} \mathbb{E}_{(x, y_1, y_2, c) \sim P}[\mathcal{L}_{\text{DPO}}(\pi_\theta)], \tag{6}
$$

where $\mathcal{Q}(\mathcal{D}) := \{P : D_{\text{KL}}(P\|\mathcal{D}) \leq \rho\}$. Using duality, the inner maximum (the objective to minimize) is approximated by:

$$\beta' \log \mathbb{E}_{(x,y_1,y_2,c)\sim\mathcal{D}} \left[ \exp\left( \frac{1}{\beta'} \mathcal{L}_{\text{DPO}}(\pi_\theta) \right) \right], \tag{7}$$

where $\beta'$ is a hyperparameter.

## D.2 Multi-armed Restless Bandits

A Multi-Armed Restless Bandit problem is characterized by N Markov Decision processes, each defined by the tuple $\{\mathcal{S}, \mathcal{A}, \mathcal{R}, \mathcal{T}, \gamma\}$. Here, $\mathcal{S}$ represents the set of states, $\mathcal{A}$ represents the set of actions which are binary in $\{0, 1\}$, $\mathcal{T} : \mathcal{S} \times \mathcal{A} \times \mathcal{S} \to [0, 1]$ represents the transition probability from current state, under an action to the next state. $\mathcal{R} : \mathcal{S} \to \mathbb{R}$ represents a reward function mapping state to a real number, and $\gamma$ represents the discounting factor. We also have a budget $K$ which represents the number of arms that can be chosen at every timestep. The objective of the Multi-Armed Restless Bandit problem then is to decide a vector of actions $\mathbf{A}^t$ representing actions $a_i^t \in \mathcal{A}$ for every arm i in time step $t$ such that we maximize long term reward under the budget constraint:

$$J(s) = \max_{\mathbf{A}\in\mathcal{A}} \left( \sum_{i=1}^{N} R_i(s_i) + \gamma \mathbb{E}[J(s') \mid s, \mathbf{A}] \right)$$

$$s.t. \sum_{i=1}^{i=N} A_i^t \leq K \ \forall \ t = 1 \text{ to } T$$

where $s_t \sim T(s_{t-1}, a_t, .)$

## D.3 Whittle Index Policy

Whittle index is a heuristic that is asymptotically optimal for planning in Multi-Armed Restless Bandits. To calculate whittle index, first we define the value function as

$$V_i^\lambda(s) = \max_{a\in\{0,1\}} Q_i(s, a_i, \lambda). \tag{8}$$

Here, $Q_i(s, a_i, \lambda)$ measures the expected discounted cumulative future reward where a reward compensation $\lambda$ is added to the reward when the passive action is taken. The Whittle index associated to the state $s_i$ is then defined as:

$$W_i(s_i) := \inf_m \{Q_i(s_i, a_i = 0, m) = Q_i(s_i, a_i = 1, m)\}.$$

The Whittle Index intuitively captures the value of taking an active action on an arm. We compute the Whittle Index using a binary search algorithm proposed by *Qian et al. 2016.*

Finally, as a policy, we choose at every timestep the beneficiaires with top-K whittle indices.

## D.4 Details on the Public Health Experiment

We apply our proposed method in the context of mobile health program run by an NGO ARMMAN [4]. Specifically, we have anonymized beneficiary listenership data from a quality-improvement study conducted by ARMMAN[2]. The data consists of registration information of beneficiaries (mothers) containing their socio-demographic features of age, income education as well as their preferences on call slot times and language. Additionally, we have information on beneficiaries' interaction with automated voice calls and live service calls made by health-workers. We model the problem of allocating live service calls as a RMAB problem. For all our experiments, we consider a sample of population with 2100 beneficiaries and $K = 210$ as the budget, representing the number of live service calls that can be made every week. Exact policy optimization for RMABs is generally known to be intractable [28]. However, Whittle index policy [34] is a popular heuristic to obtain an asymptotically optimal policy.

---

[2]The authors have been access to this restricted dataset under a data usage agreement.

To conduct our experiments in the ARMMAN real-world setting, we use a dataset of 2100 beneficiaries as a subset from a quality improvement study conducted in January 2022. Specifically, we use transition probabilities of beneficiaries estimated from historical data and use these probabilities to simulate beneficiary behaviour. Table 4 shows the beneficiary attributes we use as the set of features that LLM can use in the reward function. This includes features such as sociodemographic attributes, registration information, as well as preference on time slots of calls and on the language in which they want to receive calls.

Table 4: Features describing beneficiaries in the ARMMAN dataset

| Feature Name | Data Type |
| --- | --- |
| Enrollment gestational age | Int |
| Enrollment delivery status | Int |
| Gravidity (number of pregnancies) | Int |
| Parity (number of viable pregnancies) | Int |
| Live births count | Int |
| Days to the first call | Int |
| Ages 10–20 | Binary |
| Ages 21–30 | Binary |
| Ages 31–40 | Binary |
| Ages 41–50 | Binary |
| Ages 51–60 | Binary |
| Speaks Hindi | Binary |
| Speaks Marathi | Binary |
| Speaks Gujurati | Binary |
| Speaks Kannada | Binary |
| Education level 1/7 – illiterate | Binary |
| Education level 2/7 – 1–5th Grade Completed | Binary |
| Education level 3/7 – 6–9th Grade Completed | Binary |
| Education level 4/7 – 10th Grade Passed | Binary |
| Education level 5/7 – 12th Grade Passed | Binary |
| Education level 6/7 – Graduate | Binary |
| Education level 7/7 – Post graduate | Binary |
| Phone owner 0 (e.g., woman) | Binary |
| Phone owner 1 (e.g., husband) | Binary |
| Phone owner 2 (e.g., family) | Binary |
| To be called from 8:30am–10:30am | Binary |
| To be called from 10:30am–12:30pm | Binary |
| To be called from 12:30pm–3:30pm | Binary |
| To be called from 3:30pm–5:30pm | Binary |
| To be called from 5:30pm–7:30pm | Binary |
| To be called from 7:30pm–9:30pm | Binary |
| NGO | Binary |
| ARMMAN | Binary |
| PHC | Binary |
| Income bracket -1 (no income) | Binary |
| Income bracket 1 (0–5000) | Binary |
| Income bracket 2 (5001–10000) | Binary |
| Income bracket 3 (10001–15000) | Binary |
| Income bracket 4 (15001–20000) | Binary |
| Income bracket 5 (20001–25000) | Binary |
| Income bracket 6 (25001–30000) | Binary |
| Income bracket 7 (30000–999999) | Binary |

## D.5 RMAB Preference Data Generation

To generate the preference dataset for fine-tuning an LLM, we follow these steps:

1. We consider a set of 190 human preference commands similar to the ones proposed in [6].
2. For each of these commands, we use the DLM pipeline to generate a set of 20 relevant reward functions using prompt in Figure 2.
3. We sample a set of 50 pairs from these reward functions. For each of these pairs
   (a) solve the RMAB problem and generate trajectories for every reward function
   (b) ask an LLM Judge to decide the better reward function using the prompt in Figure 3.
4. We perform this query 10 times to estimate LLM's uncertainty in preference for every pair of chosen and rejected reward functions.

## D.6 Hyperparameters

Table 5 shows the hyperparameters used during training.

Table 5: Hyperparameter Configuration

| Description | RMAB | UltraFeedback |
|---|---|---|
| **Model Configuration** | | |
| Maximum sequence length | 2048 | 1536 |
| LoRA rank parameter | 64 | 64 |
| LoRA scaling parameter | 16 | 16 |
| LoRA dropout rate | 0.05 | 0.05 |
| Data type | bf16 | bf16 |
| **SFT Training** | | |
| Training epochs | 2 | 2 |
| learning rate | 1e-5 | 1e-6 |
| per device train batch size | 8 | 4 |
| number of devices | 4 | 4 |
| gradient accumulation steps | 1 | 2 |
| optimizer | adamw_torch | adamw_torch |
| weight decay | 0.01 | 0.01 |
| optimizer warmup steps | 100 | 100 |
| learning rate scheduler | cosine | cosine |
| **DPO Training** | | |
| training epochs | 1 | 1 |
| learning rate | 1e-5 | 1e-6 |
| per device train batch size | 4 | 2 |
| gradient accumulation steps | 4 | 4 |
| optimizer warmup steps | 100 | 100 |
| beta parameter | 0.25 | 0.25 |
| optimizer | adamw_torch | adamw_torch |
| warmup steps | 100 | 100 |
| learning rate scheduler | cosine w/min_lr=1e-6 | cosine w/min_lr=1e-7 |

# E Prompts

## LLM prompt to generate reward function

**Prompt**

Create a Python reward function for RL in phone call resource allocation to mothers in India, with the objective of prioritizing higher states and: {goal_prompt}.

The function should use 'state' (value is either 0 or 1) and features 'agent_feats' to direct the RL agent.

Here is a description of the features you may use along with the index in the 'agent_feats' array: {PROMPT_FEAT_DESC_DICT}

Your task:
1. Write a simple, single-line Python reward function. - Exclude the word 'return'. - Exclude non-standard libraries. - Format your code with []: [YOUR FUNCTION].

2. Note that HIGHER states are always preferred, so ensure the reward increases as 'state' increases.

3. Make sure the reward is always positive and increasing with state.

4. Avoid using bitwise operators. Use 'and', 'or' instead.

Here is an example to guide you: {PROMPT_REW_EXAMPLE_DICT}

Come up with a unique new reward for the specified goal: {goal_prompt}.

Figure 2: Prompt passed to the LLM to generate a reward function based on the context of problem scenario in Real World Domain.

## LLM-Judge to Decide Preferred Response

### Prompt

My goal was to create a Python reward function for RL in resource allocation, with the objective of: Focus on those with high education. I tried several reward functions for this task. Below, I have the given reward function, and the corresponding distribution of reward achieved.

Below are the reward functions I used and their corresponding reward distributions:

**Function Number 0:**    Reward Function: -agent_feats[5] -agent_feats[6]-agent_feats[7]-agent_feats[8]-agent_feats[9]-agent_feats[10]-agent_feats[11]

Reflection: '

Category: Age Ages 10-20: 121.73
Ages 21-30: 421.04
Ages 31-40: 244.49
Ages 41-50: 64.11
Ages 51-60: 10.58

Category: Income Income bracket 1 (e.g., 0-5000): 126.82
Income bracket 2 (e.g., 5001-10000): 373.62
Income bracket 3 (e.g., 10001-15000): 234.87
Income bracket 4 (e.g., 15001-20000): 77.40
Income bracket 5 (e.g., 20001-25000): 35.58
Income bracket 6 (e.g., 25001-30000): 2.58
Income bracket 7 (e.g., 30000-999999): 11.09

...
Category: Education Illiterate: 39.91
1-5th Grade Completed: 157.84
6-9th Grade Completed: 281.36
10th Grade Passed: 197.64
12th Grade Passed: 103.18
Graduate: 21.13
Post graduate: 60.89

**Function Number 1:**  Reward Function: state * agent_feats[10]
Reflection: ' Category: Age Ages 10-20: 134.22
Ages 21-30: 469.16
Ages 31-40: 270.44
Ages 41-50: 72.80
Ages 51-60: 11.96

Category: Income Income bracket 1 (e.g., 0-5000): 138.40
Income bracket 2 (e.g., 5001-10000): 414.44
Income bracket 3 (e.g., 10001-15000): 266.44
Income bracket 4 (e.g., 15001-20000): 85.33
Income bracket 5 (e.g., 20001-25000): 40.20
Income bracket 6 (e.g., 25001-30000): 2.80
Income bracket 7 (e.g., 30000-999999): 10.96
...
Category: Education Illiterate: 45.07
1-5th Grade Completed: 173.82
6-9th Grade Completed: 314.07
10th Grade Passed: 217.31
12th Grade Passed: 113.02
Graduate: 29.36
Post graduate: 65.93

Based on the above reward distributions and the given goal: Focus on those with high education., please identify the FUNCTION NUMBER of the most effective reward function. Provide your answer EXACTLY IN the following format: 'The best reward function is at number: [FUNCTION NUMBER]'.

### Output:

The best reward function is at number: 1

Figure 3: Prompt passed to the LLM to choose a reward function based on the context of problem scenario in Real World Domain, the generated reward functions and the reward distribution corresponding to every reward function.

