# OpenReview forum: "Lightweight Robust Direct Preference Optimization"
_NeurIPS.cc/2025/Workshop/Reliable_ML — NeurIPS 2025 - Reliable ML Workshop_

### Official Review · Reviewer_ZH5f · 2025-09-07
**robust framework for DPO with noisy data**

**Rating:** 10
**Confidence:** 2

**Review:**

**Strength**
1. The paper addresses a significant challenge in real-world deployment: the presence of noise in preference data. The authors clearly articulate the motivation behind their approach and propose a well-defined methodology to address this issue.

2. In addition to providing strong empirical evidence, the authors support their method with meaningful theoretical insights, which help to deepen understanding of the proposed approach.

**Weaknesses**
1. The analysis of the parameters $\rho$ and $q$ is rather limited. A more thorough investigation or sensitivity analysis would strengthen the paper’s claims and provide better guidance for practitioners.

---

### Official Review · Reviewer_wjGd · 2025-09-20
**Review: DPO-PRO: Direct Preference Optimization with Preference Robustness**

**Rating:** 7
**Confidence:** 3

**Review:**

1. **Summary.**
   This paper proposes a distributional robustification of the Direct Preference Optimization (DPO) algorithm for training LLMs with human feedback. More specifically, the authors consider pairs of annotated data $((x_1, y_1), (x_2, y_2)) \sim D$ drawn from some distribution as well as a random label $c=1$ or $c=-1$ determining preference of the pairs under a learnt reward function $R_\phi(\cdot, \cdot)$. More specifically, the authors adopt the Direct Policy Optimization model, that collapses the reward and policy learning into the minimization of a single loss function
   $\ell(x, y_1, y_2, c) = -\log \sigma\!\left(c\,\beta\left(\frac{\pi_\theta(y_1 \mid x)}{\pi_{\mathrm{ref}}(y_1 \mid x)} - \frac{\pi_\theta(x_2 \mid y_2)}{\pi_{\mathrm{ref}}(x_2 \mid y_2)}\right)\right)$.
   The main contributions of the paper are threefold: first, the authors, propose the DPO-PRO algorithm, a distributionally robust version of DPO, **defined as**

$$
\min_{\theta} \mathbb{E} [ \sup_{q:\chi^2(q\Vert p)\le \rho} \mathbb{E}_{c\sim Q}\ell(x,y_1,y_2,c) ].
$$

   Since the maximization problem is over a $\chi^2$-ball of radius $\rho$ and since $p$ is the law of a Bernoulli variable, the authors are able to compute the inner maximization in closed form.
   They, thus, obtain a closed form expression for the robustified loss function and its gradient.
   Secondly, the authors demonstrate that the DPO-PRO loss function can be thought of as a regularized DPO and provide intuition for the regularization term.
   Thirdly, the authors provide experimental results for the DPO-PRO algorithm, including an evaluation of DPO-PRO against other preference optimization algorithms (DPO, rDPO, DrDPO) using the Eurus 7B reward model.

2. **Strengths.**
   The paper is quite interesting, addressing a rather timely topic that neatly fits within the scope of the workshop -- indeed, the authors explicitly consider preference fine-tuning methods robust to distributional shifts. The closed form results are quite neat and intuitive and the authors do a good job of fleshing that out. Furthermore, the experiments seem comprehensive, although this reviewer did not check any code or data.

3. **Weaknesses / Limitations.**
   This reviewer is puzzled by various aspects of the first experiment. First, it is unclear how the “chosen response” $y_c$ is selected.
   More importantly, the results indicate that DPO-PRO outperforms all other methods at all noise levels. The authors do address this, claiming it is a result of “reward hacking”, but this reviewer, being unfamiliar with the literature, would like a more thorough explanation of this phenomenon.
   Moreover, the gap between DPO-PRO and the other methods decreases as the noise level increases, a concerning phenomenon if one is to view DPO-PRO as a robustification of DPO.

4. **Suggestions for Authors.**
   Please consider using capital letters for random variables. In expressions such as the display below line 86 is can get hard to discern between parameters, such as $\beta$ and random variables, such as $c$. Moreover, please explain the uniformly better performance of DPO-PRO in the first experiment more thoroughly as well as the decreasing gap between DPO-PRO and the other methods as noise increases. Upon satisfactorily addressing these concerns, this reviewer believes this paper will be a good fit for the workshop.

5. **Ethics.**
   The only possibly ethical concerns pertain to the proper handling of sensitive information, should that exist, in the ARMMAN dataset used in the second experiments. This reviewer urges the authors to ensure that all ethical guidelines are followed in this regard.